# MicroRNA-138-5p Targets Pro-Apoptotic Factors and Favors Neural Cell Survival: Analysis in the Injured Spinal Cord

**DOI:** 10.3390/biomedicines10071559

**Published:** 2022-06-30

**Authors:** Rodrigo M. Maza, María Asunción Barreda-Manso, David Reigada, Ágata Silván, Teresa Muñoz-Galdeano, Altea Soto, Ángela del Águila, Manuel Nieto-Díaz

**Affiliations:** 1Molecular Neuroprotection Group, Research Unit, National Hospital for Paraplegics (SESCAM), 45071 Toledo, Spain; mbarreda@sescam.jccm.es (M.A.B.-M.); dreigada@sescam.jccm.es (D.R.); agsi@lundbeck.com (Á.S.); tmunozd@sescam.jccm.es (T.M.-G.); alteas@sescam.jccm.es (A.S.); angela.delaguila@duke.edu (Á.d.Á.); 2H. Lundbeck A/S, Physiology and Symptoms, Valby, Faculty of Health and Medical Sciences, Denmark and University of Copenhagen, 2200 Copenhagen, Denmark; 3Department of Anesthesiology, Duke University Medical Center, 134 Research Dr., Durham, NC 27710, USA

**Keywords:** neuroprotection, spinal cord injury, miR-based therapies, cell death

## Abstract

The central nervous system microRNA miR-138-5p has attracted much attention in cancer research because it inhibits pro-apoptotic genes including CASP3. We hypothesize that miR-138-5p downregulation after SCI leads to overexpression of pro-apoptotic genes, sensitizing neural cells to noxious stimuli. This study aimed to identify miR-138-5p targets among pro-apoptotic genes overexpressed following SCI and to confirm that miR-138-5p modulates cell death in neural cells. Gene expression and histological analyses revealed that the drop in miR-138-5p expression after SCI is due to the massive loss of neurons and oligodendrocytes and its downregulation in neurons. Computational analyses identified 176 potential targets of miR-138-5p becoming dysregulated after SCI, including apoptotic proteins CASP-3 and CASP-7, and BAK. Reporter, RT-qPCR, and immunoblot assays in neural cell cultures confirmed that miR-138-5p targets their 3′UTRs, reduces their expression and the enzymatic activity of CASP-3 and CASP-7, and protects cells from apoptotic stimuli. Subsequent RT-qPCR and histological analyses in a rat model of SCI revealed that miR-138-5p downregulation correlates with the overexpression of its pro-apoptotic targets. Our results suggest that the downregulation of miR-138-5p after SCI may have deleterious effects on neural cells, particularly on spinal neurons.

## 1. Introduction

Injury to the spinal cord (SCI) is a leading cause of permanent disabilities with dramatic physical, societal, and medical costs (for a detailed description see [1]). SCI triggers a barrage of damaging events that spread cell death through the Central Nervous System (CNS) [1]. Many of these noxious stimuli activate apoptosis among spinal cells during the weeks that follow the injury [2,3,4,5]. The different apoptotic pathways converge in the activation of effector caspases, which are responsible for cleaving structural and functional proteins and leading to cell demise [6]. The noxious events triggered by SCI also alter gene expression in neural cells, including the overexpression of effector caspases Casp3 (coding for caspase-3 or CASP-3 protein) and Casp7 (caspase-7 or CASP-7 protein), or the pro-apoptotic mediators Bak1 (Bcl-2 homologous antagonist/killer or BAK protein), Bax, or Fas [7,8,9]. Post-transcriptional regulatory mechanisms contribute to orchestrating these gene expression changes [10,11]. Among them, studies have shown that microRNA dysregulation accompanies gene expression alterations in the injured spinal cord [12,13,14].

MicroRNAs are a class of highly conserved 20–24 nucleotide-long noncoding RNA molecules that function as post-transcriptional regulators of cell state in physiological and pathological conditions [15]. They are highly expressed in the mammalian CNS, including the spinal cord, and their dysregulation is associated with neurodegenerative, psychiatric, and developmental diseases [16,17,18]. MicroRNA dysregulation after SCI has been proposed to contribute to spreading cell death due to their targeting of key apoptotic genes. For example, downregulation of miR-29b after SCI agrees with the overexpression of its targets Bad, Bim, Noxa, and Puma [19], whereas a decrease in miR-124 levels agrees with overexpression of its target calpain-1 [20], and miR-137 downregulation agrees with calpain-2 and Casp-3 overexpression [21].

Microarray and bioinformatics analyses in a rat model of contusive SCI [14] allowed us to identify additional dysregulated microRNAs that may also contribute to activating the apoptotic pathways. Among them, here we focus on miR-138-5p, a microRNA highly expressed in the CNS that becomes significantly downregulated after SCI [12,14]. This microRNA controls the shape and size of dendritic spines in rat hippocampal neurons during development and thereby influences long-term memory [22]. Following injury, miR-138-5p participates in axon regeneration in peripheral nerves [23] and promotes neuroplasticity through the regulation of vimentin in the damaged spinal cord [24]. Outside the nervous system, miR-138-5p has attracted much attention in cancer research because it overcomes apoptotic cell death by simultaneous inhibition of multiple tumor suppressor pathways and pro-apoptotic genes, including Casp3 [25]. Therefore, we hypothesize that, in addition to the previously described effects in the CNS, miR-138-5p downregulation after SCI underlies the overexpression of apoptotic genes and sensitizes neural cells to noxious stimuli. To confirm this hypothesis, this study aims to: (a) determine which cellular changes underlie miR-138-5p downregulation during subacute SCI; (b) identify and validate miR-138-5p targets among the pro-apoptotic genes overexpressed following SCI; (c) confirm that miR-138-5p can protect neural cells from SCI insults, and (d) evaluate the association of the expression changes in miR-138-5p and its pro-apoptotic targets after SCI.

## 2. Materials and Methods

### 2.1. Surgical Procedures

We employed young adult (3 months old) female Wistar rats (RRID:RGD_13508588) weighing approximately 200 g for expression, biochemical, and histological analyses. Animals were housed in plastic cages in a temperature and humidity-controlled room and maintained on a 12:12 h reverse light/dark cycle with free access to food and water. We divided animals into two groups: one group without surgery before extraction (0 days post-injury (dpi) or control animals) and one injured group (injured). Injured animals were sacrificed at 3 and 7 dpi. We randomly allocated animals in each post-injury time (*n* = 9/dpi) or control group (*n* = 8) using the sequence generator utility of the online random number generator random.org (https://www.random.org, accessed on 20 September 2018). SCI surgery followed the methodology described in Yunta and cols. [14]. Briefly, we induced anesthesia by intraperitoneal injection of 50 mg/Kg sodium pentobarbital (Dolethal, Vetoquinol, Northamptonshire, UK) and analgesia with buprenorphine (0.03 mg/Kg Buprex; Reckitt Benckiser Pharmaceuticals Limited, Barcelona, Spain) before laminectomy at vertebral thoracic level 8 (T8) and a 200 Kilodynes contusion (IH Spinal Cord Impactor, Precision System, and Instrumentation, LLC, Sarasota, FL, USA). After surgery, we daily emptied the animal bladder until spontaneous voiding, and administered buprenorphine analgesic (0.03 mg/Kg) and enrofloxazine antibiotic (0.4 mg/Kg Baytril; Bayer AG, Barcelona, Spain) for two days. We confirmed paraplegia 2 days after the surgery using the Basso, Beattie, and Bresnahan locomotion score (BBB scale; [26]). Any animal with s BBB value above 7 was excluded from subsequent analyses.

For expression and biochemical analyses, we sacrificed animals by intraperitoneal injection of a 500 mg/Kg sodium pentobarbital overdose at the defined times (0, 3, or 7 dpi), and we extracted 1 cm long spinal cord fragments (approx. 70 mg) centered in the injury epicenter. We employed a pestle mixer (VWR Int; Radnor, PA, United States) to homogenize the tissue in 400 µL of 25 mM HEPES (4-(2-hydroxyethyl)-1-piperazineethanesulfonic acid; Sigma-Aldrich, Madrid, Spain) buffer supplemented with 1× protease inhibitor cocktail (Complete Cocktail, Roche, Madrid, Spain). We employed 70 µL of the homogenate for RNA analyses (RT-qPCR), 100 µL for caspase activity, and 250 µL for immunoblot. We confirmed the lack of effects of the homogenizing procedure on the RT-qPCR, immunoblot, and caspase activity values series of preliminary studies.

We carried out the housing, surgeries, post-operative care, behavioral tests, and sampling procedures at the Animal Facility of the Research Unit from the National Hospital for Paraplegics with the assistance of its personnel. Animal experimental procedures were approved by the National Hospital for Paraplegics’ Animal Care and Use Committee (ref#63/2010) and were in accordance with the European Communities Council Directive (2010/63/EU).

### 2.2. Histology, Fluorescent in Situ Hybridization (FISH), Immunofluorescence, and Image Analysis

We sacrificed the animals at 0, 3, and 7 dpi by intraperitoneal injection of 500 mg/Kg sodium pentobarbital (Vetoquinol, QN51AA01) immediately before transcardially perfusing them with saline and 4% paraformaldehyde (Sigma Aldrich, Macau, China) in 0.1 M phosphate buffer, PB, pH 7.4. We collected samples of spinal cord 1 cm long around the injury epicenter, immersed in 4% paraformaldehyde for 48 h at 4 °C and cryoprotected in 30% sucrose in PB (*w/v*) until they sink. Afterward, we embedded the samples in Tissue-Tek optimum cutting temperature compound (Sakura Finetek, Barcelona, Spain) and frozen at −80 °C until use. We sectioned the tissue in 20 μm transversal slices using an HM560 cryostat (Microm GmbH, Neuss, Germany) and mounted them on superfrost slides (Thermo Fisher Scientific, Madrid, Spain, cat#1014356190). Serial sections separated 300 μm covered the injured segment plus the adjacent rostral and caudal ones.

For FISH staining of miR-138-5p in the spinal cord sections, we followed protocol by Søe and cols. [27]. All solutions were prepared using autoclaved H_2_O-DEPC (diethylpyrocarbonate 1:1000 in distilled water; Sigma-Aldrich, Madrid, Spain). In brief, we thawed spinal cord sections and treated them with proteinase K for 15 min at 37 °C (40 µg/mL of proteinase K (Qiagen, Madrid, Spain) diluted in EDTA (1 mM) and NaCl (1 mM) in Tris/HCl 40 mM, pH 7.4 buffer). To avoid non-specific ionic bindings, we incubated the sections in an acetylation buffer composed by triethanolamine (1.3% (*v*/*v*)), HCl (0.06% (*v*/*v*)), and acetic anhydride (0.25% (*v*/*v*)) for 10 min at room temperature (RT). Then, we incubated the sections in hybridization buffer (1× miRCURY LNA miRNA ISH buffer; Qiagen) for 30 min at 65 °C before hybridizing them with miR-138-5p or negative control (cel-miR-67) probes (Eurogentec, Seraing, Belgium). We designed both probes following Søe and cols. [27] (see Appendix A online). We diluted probes to a final concentration of 200 nM in hybridization buffer 1×, denatured them for 4 min at 80 °C, and incubated them with the sections for 1 h at 65 °C. Then, we sequentially washed cells in 75 mM, 15 mM, and 1.5 mM saline-sodium citrate solutions (SSC; Fisher, Madrid, Spain) for 3 min at 65 °C each and a final wash in 1.5 mM SSC for 3 min at RT. Then, we incubated the sections in blocking buffer (horse serum (5%) and BSA (1%) in PBS-T-DEPC (DEPC treated PBS with 0.1% Tween 20) for 15 min at 37 °C and then with an alkaline phosphatase-conjugated sheep-anti-digoxigenin antibody (see details in Appendix A online) for 20 min at 37 °C. Finally, to detect hybridization we incubated the slides with the alkaline phosphatase subtract Vector Blue (Vector Laboratories, Burlingame, CA, USA) following the manufacturer’s protocol.

We carried out immunofluorescent staining to analyze miR-138-5p expression in neurons, oligodendrocytes, or astrocytes. Briefly, after the FISH protocol, we incubated the sections with specific cell markers for neurons (anti-NeuN), oligodendrocytes (anti-APC), or astrocytes (anti-GFAP) diluted in blocking solution overnight at 4 °C. Then we rinsed sections in PBS and incubated them for 2 h at RT with the appropriate Alexa Fluor-conjugated secondary antibodies (see Appendix A online) also diluted in blocking solution. Finally, we mounted the stained sections with a Fluorescent Mounting Medium (Agilent, Madrid, Spain) containing 1:30,000 of DAPI (4′,6-diamidino-2-fenilindol, Sigma). We employed the same protocol to stain sections with caspase 3 and caspase 7 markers (see Appendix A online) together with neuronal markers. Photographs of stained sections of the spinal cord were taken using a fully motorized Olympus IX83 microscope equipped with Cell Sense software.

To determine the number of neural cells expressing miR-138-5p, we first determined the number of neurons and oligodendrocytes present in an image using the cell counter tool of Fiji/ImageJ 1.53c [28]. In these analyses, we quantified all neurons present in the whole gray matter of each spinal cord section and the number of oligodendrocytes present in ROIs of 280 µm × 280 µm in the ventral, lateral, and dorsal white matter. Two researchers analyzed each image and only those neurons or oligodendrocytes identified by both observers were considered in the analyses. Both observers also independently evaluated microRNA staining and cells were identified as positive or negative upon agreement between both observers. We recorded the total number of neurons or oligodendrocytes in each image and the number of them stained for miR-138-5p. To analyze caspase 3 or 7 expression in neurons, we employed Fiji/ImageJ to draw the profile of ten representative neurons of each slice present in the dorsal and ventral horns and to measure caspase staining intensity in each one (7–10 sections per animal and 10 neurons per dorsal and ventral horn of each section, total = approximately 500 neurons/time and protein). To obtain the final caspase expression estimate, we subtracted the caspase intensity of negative controls (without primary caspase antibodies) from each slide.

### 2.3. Cell Culture

We cultured primary hippocampal neurons from 18 days old (E18) embryos of Wistar rats. After dissection, we dissociated hippocampi with trypsin (1×; ThermoFisher, Madrid, Spain) in Hanks’ Balanced Salt Solution (HBSS) medium without calcium and magnesium (Hyclone, GE Healthcare) supplemented with DNase (20 mg/mL; Roche) for 15 min at 37 °C. After washing-out trypsin solution with HBSS with calcium and magnesium (Hyclone, Knoxville, TN, USA), we dissociated the tissue through repeated pipetting in Minimum Essential Medium (MEM; Gibco) supplemented with 10% horse serum (HS; Fisher Scientific, Madrid, Spain). We seeded the cell suspension in plates pre-coated with 10 µg/mL poly-L-lysine (Sigma-Aldrich, Madrid, Spain). After 4 h at 37 °C and 5% CO_2_ in a cell culture incubator, we substituted the medium for Neurobasal (Gibco, ThermoFisher, Madrid, Spain) supplemented with 2% B-27 (Gibco), 1% GlutaMAX (Gibco), and 1% penicillin/streptomycin (Gibco). We kept the cultures at 5% CO_2_ and 37 °C for 4 days before subsequent experimental procedures.

In addition to primary cultures, we also employed: (a)C6 rat brain glioma cells (cat#CCL-107, ATCC; RRID:CVCL_0194) grown in RPMI-1640 medium (Gibco) supplemented with 10% fetal bovine serum (FBS; Gibco), 1% penicillin/streptomycin, and 1% GlutaMAX; (b) HEK293T human embryonic kidney cells (cat#CRL-1573, ATCC; RRID:CVCL_0045) grown in Dulbecco’s modified Eagle’s medium (DMEM; Gibco) with 10% FBS and 1% penicillin/streptomycin; (c) PC12 rat pheochromocytoma cells (cat#CCL-1721, ATCC; RRID: CVCL_0481) grown in DMEM with 10% HS, 5% FBS, 1% penicillin/streptomycin, 1 mM sodium pyruvate (Gibco), 0.075% sodium bicarbonate, 1% GlutaMAX, and 1 × non-essential amino acids (Gibco); and d) SH-SY5Y human neuroblastoma cells (cat#CRL-2266, ATCC; RRID:CVCL_0019) grown in a 1:1 mixture of MEM and Ham’s F-12 (Gibco) supplemented with 10% FBS, 1% penicillin/streptomycin, 1 mM sodium pyruvate (Gibco), and 1 × non-essential amino acids (Gibco). We cultured all cell lines in a humidified incubator at 37 °C with an atmosphere containing 5% CO_2_.

### 2.4. Computational Prediction of miR-138-5p Targets

We employed TargetScan7.1 (http://www.targetscan.org, accessed on 6 March 2020) [29], miRmap (http://mirmap.ezlab.org/, accessed on 6 March 2020) [30], miRanda (http://www.microrna.org, accessed on 6 March 2020) [31], and miRWalk2.0 (http://zmf.umm.uni-heidelberg.de/apps/zmf/mir walk2, accessed on 6 March 2020) [32] microRNA target prediction software to identify miR-138-5p response elements in rat messenger RNAs (mRNA). We employed the GEO2R web tool [33] to identify those genes that become dysregulated in rat spinal cord during the first 10 dpi according to high throughput gene expression data stored in the GEO datasets (https://www.ncbi.nlm.nih.gov/gds, accessed on 10 March 2020) references GSE464 (1, 3, and 7 dpi, [34]) and GSE69334 (3 and 10 dpi, [35]). To identify genes related to cell death, we annotated the obtained list of dysregulated genes using the Functional Annotation tool (default criteria) of the Database for Annotation, Visualization, and Integrated Discovery (DAVID, http://david.abcc.ncifcrf.gov/, accessed on 13 March 2020) [36]. We explored validated microRNA-target interactions using miRTarBase 8.0 database (https://mirtarbase.cuhk.edu.cn/~miRTarBase/miRTarBase_2019/php/index.php, (accessed on 30 July 2021). To evaluate the stability of the target-miR-138-5p binding and the accessibility (i.e., how likely a region in an mRNA sequence is accessible for a microRNA to bind) of the mRNA secondary structure, we employed the following tools: (1) the mFold software (http://unafold.rna.albany.edu, accessed on 7 September 2021) [37], which we used to calculate the free energy (ΔG) of the predicted microRNA binding site and the 100 nucleotides flanking its 5′ and 3′ sides in the rat’s 3′UTR (3′ untranslated region) of the target genes; (2) the PITA software (https://genie.weizmann.ac.il/, accessed on 8 September 2021) [38], setting the criterion of ΔΔG  ≤  −10 kcal/mol and other parameters left to default. We calculated the accessibility of the miR-138-5p target site (ΔΔG) from the formula ΔΔG = ΔGduplex − ΔGopen, being ΔGopen, the energy required to open the target mRNA secondary structure and ΔGduplex, the energy gained by the microRNA binding; (3) the miRmap prediction program (https://mirmap.ezlab.org/, accessed on 6 March 2020) [30], which computes the minimum free energy of this duplex, calculated from the Vienna RNA secondary structure library; and (4) the miRWalk3.0 prediction program (http://mirwalk.umm.uni-heidelberg.de/, accessed on 6 March 2020) [32], which calculates the RNA-duplex energy (ViennaRNA package) [39].

### 2.5. Microarray Data

RNA preparation, hybridization, staining, and scanning of the microRNA array are described in Yunta and cols. [14]. The resulting expression values were downloaded from the GEO database (http://www.ncbi.nlm.nih.gov/geo/, accessed on 1 July 2011) under accession number GSE19890.

### 2.6. RT-qPCR Analysis

We employed miRNeasy Kit (Qiagen) for isolation and purification of total RNA, including microRNAs, from rat spinal cord, as well as from C6, HEK293T, PC12, and SH-SY5Y cell lines. Total RNA concentration and purity (260/280 and 260/230 ratios) were estimated with a NanoDrop ND-1000 spectrophotometer (Thermo Scientific). Only samples with 260/280 ratios between 1.8 and 2.2 were employed.

To determine miR-138-5p expression, we reverse-transcribed and amplified 10 ng of total RNA using TaqMan microRNA gene expression assay (TaqMan^®^ MicroRNA assay #002284, Applied Biosystems, Whaltman, MA, USA) following manufacturer’s protocols. We used U6 small nuclear RNA as an internal control (TaqMan^®^ MicroRNA assay #001973, Applied Biosystems). For mRNA detection of Casp3, Casp7, and Bak1 transcripts, we synthesized cDNA from 1 µg of total RNA subjected to random reverse transcription using Moloney Murine Leukemia Virus reverse transcriptase (M-MLV-RT; Invitrogen, Thermo Fisher, Madrid, Spain) and random primers (Roche). Then we evaluated the gene expression levels using TaqMan Gene Expression Assays (Applied Biosystems) for Casp3 (#00563962), Casp7 (#01410847), and Bak1 (#01429084), employing 18S ribosomal RNA (#4333760) as a housekeeping gene. We measured the abundance of the microRNA and mRNAs of interest in a thermocycler ABI Prism 7900 fast (Applied Biosystems) for 40 cycles of two steps: 15 s at 95 °C plus 1 min at 60 °C using the 2^−ΔΔCt^ method [40]. Briefly, we estimated the difference (ΔCt) between the cycle threshold of the target mRNA or microRNA and their respective endogenous loading controls (U6 for miR-138-5p and 18S for target genes) together with its associated variance using the standard propagation of error method from Headrick [41]. Then, we compared gene expression at different times post-injury relative to the value from non-injured animals (0 dpi) using the ΔΔCt method with the correspondent fold change (2^−ΔΔCt^), and the 95% confidence interval (CI).

### 2.7. Cell Death Assays

#### 2.7.1. Calcein/Propidium Iodide Assay

We seeded 20,000 hippocampal neurons per well in 48-well plates pre-coated with 10 µg/mL poly-L-lysine. Four days later, we transfected them with either 50 nM miR-138-5p (miRIDIAN cat#C-320369-05, Dharmacon, Lafayette, CO, USA) or cel-miR-67 negative control (miRIDIAN microRNA mimic negative control#1 cat#CN-001000-01; Dharmacon) mimics using a rabies virus glycoprotein-PEG functionalized 10 Kdalton unbranched polyethylenimine (PEI10K-RVG). This is an optimized modification of the RVG-disulfide-linked PEI nanocarrier developed by Hwang and cols. [42] for the in vivo transfection of microRNA to the CNS, which we will describe in detail elsewhere (preliminary results are described in M. Maza and cols. [43]). After 24 h, we stimulated cells overnight with 15 mM L-glutamic acid (LGA; Sigma-Aldrich) before incubating them with 2.5 µM calcein-AM (Sigma-Aldrich) and 0.4 µg/mL propidium iodide (PI; Sigma-Aldrich) in warm PBS supplemented with 10% FBS medium for 30 min at 37 °C protected from light. We took photographs of the cells using an epifluorescence microscope (DMIL LED, Leica Microsystem GmbH, Wetzlar, Germany) with a 20× microscope lens, coupled to a Leica DFC 3000 G camera. We used the ImageJ software (National Institutes of Health, NIH) to process and analyze the images. Calcein-AM labels viable cells, whereas PI gains access only to cells with plasma membrane damage and accumulates in the nucleus. We estimated cell death from the percentage of calcein-AM or PI-stained cells related to the total neuron number.

#### 2.7.2. Terminal Deoxynucleotidyl Transferase-Mediated Deoxyuridine Triphosphate (dUTP) Nick End-Labeling (TUNEL) Assay

We identified DNA damage in dying neurons using TUNEL label mix (Merck). We seeded 50,000 hippocampal neurons per well in 24-wells plates with coverslips pre-coated with 50 µg/mL poly-L-lysine. Four days later we transfected the neurons with either 50 nM miR-138-5p or cel-miR-67 negative control mimics using PEI10K-RVG to vehicle the RNA molecules. After 24 h, we stimulated a set of wells overnight with 15 mM LGA. Then, we fixed neuron cultures with paraformaldehyde 4% for 20 min at RT, rinsed coverslips with PBS, and incubated them with permeabilization buffer (0.1% Triton X-100 (Sigma-Aldrich) and 0.1% de sodium citrate (Sigma-Aldrich) for 2 min on ice. Afterward, we blocked unspecific binding with a solution of 3% BSA (Sigma-Aldrich) and 0.2% Triton X-100 in PBS for 1 h at RT before incubating the cells with the neuronal marker mouse anti-beta-III-tubulin antibody (1:500 in blocking solution; see antibody details in Appendix A online) overnight at 4 °C. Afterward, we rinsed coverslips with PBS and labeled the cells with TUNEL reaction mixture (Merck Millipore, Madrid, Spain) according to the manufacturer’s protocol and a fluorescent Alexa 594-conjugated goat anti-mouse IgG2a secondary antibody. After further washes with PBS, we mounted the coverslips on glass slides employing Fluorescence Mounting Medium (Thermo Scientific) with DAPI (1:30,000). We took images using an epifluorescence microscope (DM5000B, Leica Microsystem GmbH) with a 40× microscope lens, coupled to a Leica DFC 350 FXR camera. We used ImageJ software to process and analyze the images. We estimated cell death as the percentage of TUNEL-stained neurons related to the total neuron number.

#### 2.7.3. MTT Assay

We seeded 20,000 C6 cells per well overnight in transparent 96-well plates and transfected them with either miR-138-5p or cel-miR-67 negative control mimics using Dharmafect 1.0. After 24 h, we stimulated cells overnight with either 0.3 μM staurosporine (STS; Sigma-Aldrich), 127.5 μM etoposide (ETO; Sigma-Aldrich), or 1 mM LGA. Then, we incubated cell cultures for 3 h with 3-(4,5-dimethylthiazol-2-yl)-2,5-diphenyltetrazolium bromide (MTT) diluted in cell culture medium to a final concentration of 0.5 mg/mL. We dissolved generated formazan crystals by the addition of 100 μL of HCl:Isopropanol solution (1:500, Merck Millipore and Sigma Aldrich respectively) to each well followed by measurement of the absorbance at 570 nm (and 690 nm for background estimation) in a Tecan Infinite M200 plate reader.

#### 2.7.4. Quantification of Cell Death Using Flow Cytometry

For cell death analysis, we seeded 40,000 C6 cells per well in 24-well plates in duplicates for each transfection condition (miR-138-5p or cel-miR-67 negative control mimics). When reaching 50% confluence, we transfected the microRNA mimics using Dharmafect 1.0 and, 24 h later, we exposed cells to 127.5 μM ETO overnight. After detachment with 250 mM EDTA (ethylenediaminetetraacetic acid, Sigma-Aldrich), we evaluated apoptosis by Annexin V/SYTOX staining using the DY634 Annexin V Apoptosis Detection Kit (Immunostep, Salamanca, Spain) and SYTOX^™^ Blue Nucleic Acid Stain (ThermoFisher Scientific) according to the manufacturer’s instructions. We used dot plots showing pulse width versus area to distinguish between single cells and aggregates. We collected a total of 10,000 gated single events using the FACS Canto II flow cytometry (BD Biosciences, Franklin Lakes, NJ, USA) and the FACSDiva 6.1 software (BD Biosciences) and analyzed them with FlowJo Software (Celeza GmbH, Olten, Switzerland) to determine the percentage of population stained with each dye.

### 2.8. Dual Luciferase Reporter Gene Construction and 3′UTR Luciferase Reporter Assays

We amplified the complete 3′UTRs containing the predicted binding sites for miR-138-5p of rat’s Casp3 (position 1: nucleotides (nt) 709-715 and position 2: nt 759–766 of the 3′UTR from reference sequence NM_012922.2), Casp 7 (nt 847–853 of the 3′UTR from NM_022260.3), and Bak1 (nt 799–805 of the 3′UTR from NM_053812.2), by PCR using specific primers (Appendix A online; Sigma Aldrich). We subcloned the resulting products into the pGEM-T Easy vector (Promega, Madrid, Spain) and then we subcloned them at the SacI or NheI and SalI sites downstream of the firefly luciferase reporter gene of pmiRGLO (Promega). We also subcloned the 3′UTR of Fadd (Fas-associated protein with death domain) to be employed as a negative reporter construct. Site-directed mutagenesis for the 3′UTR constructs was generated by PCR using the 3′UTR-mut primers (Appendix A), PfuI polymerase (Thermo Scientific), and the wild-type 3′UTRs constructs subcloned into pGEM-T Easy vector as a template, following by the DpnI endonuclease restriction digestion and transformation of *E.coli* supercompetent cells. We inserted the 3′UTR mutated into pmiRGLO between the SacI or NheI and SalI sites downstream of the firefly luciferase reporter gene of pmiRGLO. We grew C6 and HEK293T cell lines, chosen based on their detectable levels of pro-CASP-3, pro-CASP-7, and BAK expression (see Appendix A online), to 70% confluence in white 96-well plates. Then, we co-transfected cells with either 50 nM miR-138-5p or 50 nM cel-miR-67 negative control mimics and 200 ng/well of pmiRGLO containing each wild-type or mutant 3′UTR of the pro-apoptotic genes (or pmiRGLO empty as control) employing DharmaFECT Duo Transfection Reagent (Dharmacon). We measured Firefly and Renilla luciferase activities 24 h later with a plate reader (Infinite M200, Tecan) employing the Dual-GLO luciferase assay system (Promega) according to the manufacturer’s protocol.

### 2.9. Immunoblot Assay

We analyzed pro-CASP-3 and pro-CASP-7, and BAK protein levels using standard immunoblot procedures. Briefly, we incubated C6 cell line lysates and spinal cord homogenates in radioimmunoprecipitation assay lysis buffer (RIPA, Sigma-Aldrich) supplemented with a complete EDTA-free protease inhibitor cocktail (Roche). We mildly sonicated (3 cycles of 20 s at 70% power using a Sonopuls sonicator, Bandelin, Berlin, Germany) tissue samples before clearing by centrifugation (14,000× *g*, 10 min at 4 °C). We determined protein content using the bicinchoninic acid method according to manufacturer’s protocols (BCA protein assay kit, ThermoFisher Scientific). We resolved a total of 50 µg of protein by SDS-PAGE, then electrophoretically transferred to a 0.2 µm polyvinylidene difluoride membrane (PVDF; Immobilon, Merck Millipore), and probed with antibodies against CASP-3, CASP-7, or BAK (see details on antibodies in Appendix A online) according to the manufacturer’s protocol. Beta-tubulin antibody was used as loading control. All antibodies are validated by the Cell Signalling Technology company. After incubation with the primary antibody, we washed membranes with TBS-Tween20 (Sigma Aldrich) and incubated them with a horseradish peroxidase (HRP)-conjugated goat anti-rabbit secondary antibody or an HRP-conjugated goat anti-mouse secondary antibody for 1 h at RT. Finally, we developed the HRP signal using the SuperSignal West Pico Chemiluminescent detection system (Pierce, ThermoFisher Scientific) and measured using ImageScanner III and LabScan v6.0 software (GE Healthcare Bio-Sciences AB, Uppsala, Sweden) using default settings.

### 2.10. Measurement of Caspase-3/7 Activity

We employed the luminescent Caspase-Glo^®^ 3/7 assay kit (Promega) to analyze the bulk activity of effector caspases-3/7 in cell cultures and spinal cord samples. For cell cultures, we seeded 20,000 C6 cells per well in white 96-well plates. After 24 h, we transfected cells with either miR-138- or cel-miR-67 negative control mimics employing Dharmafect 1.0 (Dharmacon). The next day, we induced cell death by treating cells with 0.3 μM STS or 127.5 μM ETO overnight. We assessed effector caspase activity 24 h after according to the manufacturer’s instructions. For spinal cord samples, we cleared 100 µL of spinal cord homogenate by centrifugation (14,000× *g* for 10 min at 4 °C) and we determined the protein content by the BCA method. We mixed 25 µL of the sample at 30 µg/mL protein concentration with 25 µL of Caspase-Glo^®^ 3/7 assay. We recorded luminescence derived from enzymatic activity every 30 s during the first 5 min and then every 5 min during the next hour in an Infinite M200 plate reader. We used the values of maximum luminescence emission in each experiment (usually the 15 min time point) in data analyses and plotting.

To analyze effector caspase activity in transfected hippocampal neurons we employed the CellEvent^™^ caspase-3/7 green detection assay (ThermoFisher Scientific) that allows analyzing activity in individual cells. Briefly, we seeded 20,000 hippocampal neurons per well in 48-well plates pre-coated with 10 µg/mL poly-L-lysine and, 4 days later, we transfected them with 50 nM of either miR-138-5p or cel-miR-67 negative control mimics (Dharmacon) using PEI10K-RVG. After 24 h, we stimulated cells overnight with 15 mM LGA. We assessed effector caspase activity 24 h later by incubation in 2.5 μM of the assay reagent in warm PBS supplemented with 10% FBS medium for 30 min at 37 °C protected from light. We took photographs of the cells using an epifluorescence microscope (DMIL LED) with a 20× microscope lens, coupled to a Leica DFC 3000 G camera. We used ImageJ software to process and analyze the images. We estimated Caspase-3/7 activity as the percentage of Caspase-stained neurons after miR-138-5p transfection normalized to the percentage of stained neurons after transfection with the negative control microRNA.

### 2.11. Data Analysis

Statistical significance of the treatment effects was tested using paired or unpaired Student’s t and ANOVA (followed by Tukey post hoc test for pairwise comparisons) depending on the characteristics of the data. Normality and homoscedasticity of the data were assessed using Shapiro–Wilkinson and Bartlett tests, respectively. The Kruskal–Wallis test followed by the Conover–Iman post hoc test were employed to substitute ANOVA and Tukey post hoc test when data were identified as non-parametric. Additionally, linear models were fitted in some analyses (vg. histological analysis) and compared with ANOVA to determine the best fit. The monotonic association between the expression levels of miR-138-5p and its targets was explored using the Spearman correlation coefficient. Most in vitro and all in vivo analyses were carried out according to a randomized block design, incorporating the experiment as a block factor in the statistical analyses. Data are expressed as mean ± SEM or mean ± SD as indicated in figure legends. Statistical analyses and graphic representations were conducted using Prism Software 5 (GraphPad Software Inc.) or R version 3.4.3 (https://www.R-project.org/, accessed on 18 March 2022) [44]. Differences were considered statistically significant when the *p*-value was below 0.05.

All raw and processed data plus the statistical analyses are available as Appendix A at the Open Science Framework repository (OSF: https://osf.io/yqn58/, accessed on 24 May 2022).

## 3. Results

### 3.1. MiR-138-5p Becomes Downregulated among Spinal Neurons during Subacute SCI

Previous reports have shown that miR-138-5p is highly expressed in the CNS, particularly among brain neurons [45,46,47]. To characterize miR-138-5p expression among spinal cord cells, we analyzed its expression through fluorescent in situ hybridization (FISH) of spine sections co-stained with cell-type-specific markers (images are available at OSF: https://osf.io/yqn58/, accessed on 24 May 2022). The distribution of miR-138-5p staining reveals that this microRNA is mostly expressed throughout the gray matter and, to a lesser degree, in the white matter (Figure 1A). Quantification of miR-138-5p positive neurons co-stained with NeuN marker reveals that it is expressed in most neurons throughout the gray matter except in the Rexed laminae I to III of the dorsal horn, where positive neurons represent less than 25% of the present neurons (Figure 1E). MiR-138-5p staining is also detected in oligodendrocytes (APC-positive cells), but the fluorescent labelling was more subtle or even absent compared with the neuronal FISH labelling (Figure 1C,D). No staining was detectable among astrocytes (GFAP-positive cells; Figure 1B).

Murine models of SCI have shown that miR-138-5p becomes downregulated in the spinal cord during the days that follow the injury [12,14]. In agreement, a reanalysis of microarray data from Yunta and cols. [14] reveals a decrease in miR-138-5p expression 3 and 7 dpi (significantly at 7 dpi *p* < 0.05, see Figure 2A). RT-qPCR analyses of rat spinal cord samples confirm the significant downregulation of miR-138-5p at 3 and 7 dpi (*p* < 0.05, see Figure 2B). To identify the spinal cells responsible for these expression changes, we quantified the number of neurons and oligodendrocytes expressing miR-138-5p after SCI. Analysis of the resulting data using linear models reveals that nearly 50% of the oligodendrocytes are miR-138-5p positive before injury and that this fraction remains almost unaltered after injury despite the extensive oligodendroglial losses that occur (F_1,61_ = 310.9, *p* < 0.001, see Figure 2C). On the contrary, the behavior of the neurons is more complex and the fraction of miR-138-5p positive neurons varies after injury (F_3,31_ = 38.86, *p* < 0.001, see Figure 2D). Compared to the undamaged spinal cord, at 3 dpi, we observed a reduction in the number of miR-138-5p positive neurons that is associated with the reduction in the total number of neurons, broadly preserving the fraction of miR-138-5p positive neurons (58% in the naive spinal cord vs. 50% at 3 dpi, Figure 2E). At 7 dpi, the number of miR-138-5p positive neurons keeps falling even though the total number of neurons remains unchanged, leading to a marked reduction in the percentage of miR-138-5p positive neurons recorded in the naïve or 3 dpi spinal cords (21%, see Figure 2E). In summary, present results indicate that injury reduces miR-138-5p expression in the damaged spinal cord due to the loss of miR-138-5p-expressing cells (neurons and oligodendrocytes) during the first days after injury as well as to a marked downregulation of this microRNA in neurons that is observed 7 dpi.

### 3.2. MiR-138-5p Protects Neurons In Vitro

Histological results indicate that injury leads to a marked downregulation of miR-138-5p in neurons 7 days after injury. To evaluate whether expression changes in miR-138-5p affect neuronal survival, we transfected hippocampal neuron cultures with either miR-138-5p or negative control mimics and stimulated them with 15 mM LGA, simulating the excitotoxic conditions observed during acute SCI. As shown in Figure 3A, neuronal death induced by stimulation with LGA (% of PI-stained cells; *p* < 0.01) is partially reversed by the overexpression of miR-138-5p, but not by the negative control (miR-138-5p mimic = 30.30% ± 4.41 PI cells, negative control = 42.31% ± 6.69, close to statistical significance, *p* = 0.055). A similar effect was observed using a calcein assay (miR-138-5p mimic = 87.56% ± 2.91 calcein-AM stained cells vs. negative control = 80.21% ± 4.66; *p* < 0.05; see Appendix A online). We further confirmed miR-138-5p neuroprotection using the TUNEL assay. Hippocampal neuron cultures transfected with the negative control mimic and stimulated with 15 mM LGA significantly increase the percentage of TUNEL-stained neurons relative to unstimulated cells (LGA = 39.64% ± 9.96, unstimulated cells = 13.92% ± 6.16; *p* < 0.001; see Figure 3B). On the contrary, transfection with miR-138-5p mimic followed by LGA treatment significantly reduces the percentage of death cells almost back to unstimulated values (miR-138-5p mimic = 20.59% ± 4.55, negative control microRNA = 39.64% ± 9.96; *p* < 0.01; Figure 3B). Altogether, these results indicate that overexpression of miR-138-5p favors neuron survival in excitotoxic conditions.

### 3.3. Predicting Novel Targets of miR-138-5p among the Genes Dysregulated after SCI

In order to identify miR-138-5p targets that may be responsible for its neuroprotective activity, we explored how miR-138-5p dysregulation affects the expression of apoptotic cell death-related genes after SCI. To do so, we initially carried out an in silico analysis to predict potential targets of miR-138-5p. Since the various available programs can yield rather different predictions, we combined TargetScan 7.1, miRWalk 2.0, miRmap, and miRanda programs to search for rat miR-138-5p gene targets (Figure 4A). MiRWalk2.0 (green) identifies a total of 1743 genes as predicted targets of miR-138-5p, whereas TargetScan (blue), miRmap (red), and miRanda (yellow) listed 2989, 2092, and 1159 genes, respectively. A total of 209 common genes are identified by the four prediction programs (the full list is available in the Appendix A online). A comparison of this list with those genes differentially expressed after SCI according to GEO datasets GSE464 (1, 3, and 7 dpi) and GSE69334 (3 and 10 dpi) reveals that 176 out of the 209 predicted miR-138-5p targets became dysregulated during the first days after injury (Figure 4A, see details in the Appendix A online). Functional annotation of these 176 genes using DAVID bioinformatics resources identifies Gene Ontology terms for Biological Processes involving the apoptotic process or cell cycle including GO:0051402 (neuron apoptotic process), GO:2001238 (positive regulation of extrinsic apoptotic signaling pathway), GO:0097190 (apoptotic signaling pathway), GO:0006915 (apoptotic process), GO:0043066 (negative regulation of apoptotic process), GO:0043065 (positive regulation of apoptotic process), and GO:0007049 (cell cycle) (the full list is available in the Appendix A online). Among the miR-138-5p putative targets involved in apoptotic processes, we selected Casp3, Casp7, and Bak1 genes for detailed analysis, attending to their role in apoptotic cell death following SCI as well as to their values in the algorithm prediction scores (Figure 4A). According to the employed prediction programs, rat Casp3 mRNA has two binding sites for miR-138-5p on its 3′UTR, whereas Bak1 and Casp7 mRNA present only one site. The alignment of these putative sites in rat, mouse, and human sequences demonstrates the evolutionary conservation of Casp3 (site 2), Casp7, and Bak1 sites among mammalian species. No such conservation is observed for site 1 of Casp3 (Figure 4B). Exploration of the miRTarBase8.0 database (last accessed on 30 July 2021) reveals that only Casp3 had been already experimentally validated as a target of human miR-138-5p.

Analyses of target site accessibility of the mRNA secondary structure further support miR-138-5p targeting on the three predicted apoptotic targets (Figure 4C). Free energy (ΔG) values computed using mFold indicate that the miR-138-5p binding sites in the sequences of Casp3, Casp7, and Bak1 have a lower ΔG (mean ΔGCasp3 = −12.7 kcal/mol; mean ΔGCasp7 = −11.79 kcal/mol; mean ΔGBak1 = −17.17 kcal/mol) than their corresponding 5′ and 3′ flanking regions, suggesting that miR-138-5p binding is particularly stable in these sites. Similarly, target site accessibility calculated using PITA software shows that the predicted interaction sites correspond to highly stable microRNA-mRNA duplexes formed by 8 mer (ΔGduplex Casp3 = −21.8 kcal/mol; ΔGduplex Casp7 = −29.1 kcal/mol; and ΔGduplex Bak1 = −22.7 kcal/mol) with low overall energy requirements (ΔΔGCasp3  = −18.2 kcal/mol; ΔΔGCasp7 = −18.77 kcal/mol; and ΔΔGBak1 = −12.01 kcal/mol). MiRmap predictions on the minimum free energy of the microRNA-mRNA duplexes also support a stable structure of the duplex of miR-138-5p and the three mRNA targets (ΔGbindingCasp3 = −22.65 kcal/mol, ΔGbindingCasp7 = −23.63 kcal/mol, and ΔGbindingBak1 = −19.93 kcal/mol). The minimal folding energy of the RNA-duplex calculated using miRWalk 3.0 software renders similar results (ΔGbindingCasp3 = −22.0 kcal/mol, ΔGbindingCasp7 = −28.6 kcal/mol and ΔGbindingBak1 = −21.1 kcal/mol). Moreover, miRWalk3.0 shows as well a high accessibility of the duplexes of miR-138-5p with Casp3 (accessibility: site1 = 0.21, site2 = 0.01), Casp7 (accessibility = 0.007), and Bak1 (accessibility = 0.002). Taken together, bioinformatics approaches agree that miR-138-5p has potential sites in the sequences of Casp3, Casp7, and Bak1, and therefore miR-138-5p can play a biologically relevant role in regulating their expression to protect neurons against deleterious stimuli such as LGA.

### 3.4. Reporter Assays Validate miR-138-5p Targeting on Casp3, Casp7, and Bak1 3′UTRs

We employed reporter assays to experimentally validate the three miR-138-5p apoptotic targets predicted in silico. We selected the human HEK293T and the rat C6 cell lines for these analyses due to their detectable levels of pro-CASP-3, pro-CASP-7, and BAK expression (see Appendix A online) and their low to moderate endogenous expression of miR-138-5p (Appendix A online).

As shown in Figure 5A, transfection of miR-138-5p mimics in C6 cells significantly reduces luciferase activity (Firefly/Renilla ratio) of the constructs bearing the 3′UTRs of Casp3 (mean reduction ± SD = 23.17% ± 11.4, *n* = 5, t_4_ = 6.429, *p* < 0.01), Casp7 (17.88% ± 10.584, *n* = 4, t_3_ = 5.506 *p* < 0.05), or Bak1 (16.89% ± 12.58, *n* = 4, t_3_ = 4.36, *p* < 0.05), but not with subcloned Fadd’s 3′UTR (−6.5% ± 26.9, *n* = 3, t_2_ = 0.643, *p* = 0.17) compared to the values obtained after co-transfection with the negative control mimic. Luciferase activity is also significantly reduced, although to a lesser extent, when HEK293T cells were co-transfected with the miR-138-5p mimic and the luciferase construct of Casp3 (20% ± 20.31, *n* = 5, t_4_ = 3.113, *p* < 0.05) (Appendix A online). Though not statistically significant, a similar trend is observed for the constructs of Casp7 (2.33% ± 6.9, *n* = 3, t_2_ = 1.28, *p* = 0.32) and Bak1 (6.76% ± 6.34, *n* = 3, t_2_ = 4.056, *p* = 0.055). As for C6 cells, the expression of Fadd construct remains unaffected by the transfection of the miR-138-5p mimic (−1.58% ± 27.6, *n* = 3, t_2_ = 1.28, *p* = 0.86) (Figure 3B). Altogether, results from these reporter assays confirm that miR-138-5p regulates the 3′UTRs of Casp3, Casp7, and Bak1.

Conversely, transfection with miR-138-5p mimics does not cause any reduction in luciferase activity in C6 cell cultures transfected with luciferase constructs bearing the mutated 3′UTRs (Casp3-mut = 4.68% ± 21.77, *n* = 4, t_3_ = 0.430, *p* = 0.652; Casp7-mut = 12.94% ± 30.99, *n* = 4, t_3_ = 0.835, *p* = 0.767, and Bak1-mut = −9.15% ± 16.78, *n* = 4, t_3_ = −1.091, *p* = 0.177; Figure 5B), which confirms the specificity of miR-138-5p regulation on the predicted binding sites of the three target genes. Transfection of miR-138-5p mimic does not reduce luciferase activity of the empty construct (Ø bar), confirming that miR-138-5p has no effect on the luciferase reporter construct that may bias the results. Conversely, transfection of the subcloned Casp3, Casp7, and Bak1 3′UTR luciferase constructs (black bars), as well as its co-transfection with the negative control miRNA mimic (gray bars), in C6 cell cultures leads to a significant reduction in the luciferase activity (Figure 5A). A similar reduction is observed in human HEK293T cell cultures, except when co-transfecting the Casp3 3′UTR construct (Appendix A online), which suggests that the expression of the three genes is subjected to endogenous regulation in both cell lines.

### 3.5. MiR-138-5p Reduces the Gene and Protein Expression of Caspase 3, Caspase 7, and BAK

To evaluate whether miR-138-5p modulates the cellular levels of Caspase 3, Caspase 7, and BAK, we employed RT-qPCR and immunoblot assays to measure the effect of transfecting miR-138-5p mimic on their transcript and protein expressions. Comparison of the mRNA levels of Casp3, Casp7, and Bak1 in C6 cell cultures transiently transfected with miR-138-5p or negative mimics reveals that miR-138-5p mimic significantly decreases gene expression of Casp3 (*p* < 0.05), Casp7 (*p* < 0.01), and Bak1 (*p* < 0.05) (Figure 6A). In agreement with the mRNA levels, transfection of miR-138-5p leads to a significant downregulation of the endogenous protein levels of pro-CASP-3 (56% ± 21, *n* = 6, t_5_ = 5.06, *p* < 0.01), pro-CASP-7 (48% ± 19, *n* = 4, t_3_ = 5.31, *p* < 0.05), and BAK (54% ± 22, *n* = 6, t_5_ = 5.02, *p* < 0.01) relative to transfection with negative control (Figure 6B). As a whole, our results validate that miR-138-5p downregulates the expression of its pro-apoptotic targets Casp-3, Casp-7, and BAK in C6 cells at both mRNA and protein levels.

### 3.6. MiR-138-5p Attenuates Caspase-Dependent Apoptosis

Since Caspase 3, Caspase 7, and BAK actively participate in apoptosis, we investigated whether their regulation by miR-138-5p also affects their activity in this cell death process. We focused these analyses on the effector CASP-3 and CASP-7, leaving BAK unanalyzed because its apoptotic activity ultimately relies on (and is reflected by) the cleavage and activation of both effector proteases [48]. Before carrying out these functional studies, we first extended the immunoblot analysis of pro-CASP-3 and pro-CASP-7 to examine their regulation by miR-138-5p in C6 cells under apoptotic conditions. It is well established that apoptotic stimulation leads to the proteolytic cleavage of the effector caspases and, concomitant, to the reduction in the levels of their proforms. Accordingly, stimulation of control C6 cells (i.e., transfected with negative control mimic) with 350 µM ETO reduces the level of pro-CASP-3 and pro-CASP-7 relative to unstimulated cells (pro-CASP-3 reduction = 52.37% ± 18.18, *n* = 5; pro-CASP-7 reduction = 75.43% ± 27.61, *n* = 4; see Figure 7A). Similar reductions in pro-caspase expression are observed after transfection with miR-138-5p mimic without ETO stimulation (pro-CASP-3 reduction = 49.38% ± 22.99; pro-CASP-7 reduction = 71.16% ± 16.90; see Figure 7A) confirming the regulatory effects of miR-138-5p shown in Figure 6. In both pro-caspases, the decrease in expression induced by miR-138-5p is additive to the reduction induced by ETO stimulation leading to significantly lower expression values when both stimuli are coupled (pro-CASP-3 reduction = 88.24% ± 14.22; pro-CASP-7 reduction = 87.92% ± 15.38; see Figure 7A). These results indicate that the effects of ETO stimulation and miR-138 overexpression are independent and additive. Overall, immunoblot analyses indicate that miR-138-5p overexpression reduces caspase expression in C6 cells under apoptotic stimulation, which ultimately should result in the diminished activity of these proteases. To verify this hypothesis, we quantified the enzymatic activity of effector caspase-3/7 in C6 cell cultures transfected with either miR-138-5p or negative control mimics and exposed to ETO and STS apoptotic stimulations. As shown in Figure 7B, the enzymatic assay reveals that the activity of the effector caspase-3/7 was slightly and not significantly lower when C6 cell cultures were transfected with the miR-138-5p mimic in basal conditions (caspase activity reduction = 8.37% ± 2.47, *n* = 4, *p* = 0.135) and upon apoptotic stimulation with either 127.5 μM ETO (reduction = 17.44% ± 3.68, *p* < 0.01) or 0.3 μM STS (reduction = 19.42% ± 11.32, *p* < 0.01). These results confirm that transfection with the miR-138-5p mimic reduces the amount of available pro-CASP-3 and pro-CASP-7 to become cleaved and, therefore, active. We also tested whether transfection with miR-138-5p mimic also reduces caspase-3/7 activity in cultures of hippocampal neurons. In these cells, LGA stimulation leads to a significant increase in caspase-3/7 activity (F_1,6_ = 6.420, *n* = 3, *p* < 0.05), which is partially counteracted by miR-138-5p transfection (F_1,6_ = 12.966, *p* < 0.05), in agreement with results in C6 cells (Figure 7C). Finally, we evaluated whether miR-138-5p overexpression protects neural cells from apoptotic stimuli. To test this possibility, C6 cell cultures transfected for 24 h with either negative control or miR-138-5p mimics were stimulated with 1 mM LGA, 127.5 μM ETO, or 0.3 μM STS, and their viability was analyzed using the MTT assay. As illustrated in Figure 7D, transfection of miR-138-5p mimic significantly increases cell viability (F_1_ = 12.491, *n* = 3, *p* < 0.01) irrespective of whether cells were stimulated with LGA (MTT value increase = 71.81% ± 9.00), ETO (increase = 113% ± 27.69), or STS (increase = 131.49% ± 31.50). We further assayed cell viability using flow cytometry in C6 cultures transfected with either miR-138-5p or negative control mimics and stimulated with 127.5 μM ETO (Figure 7E). In agreement with the MTT results, overexpression of miR-138-5p significantly increases cell viability following treatment with ETO (viable cells increase = 32.02% ± 21.47, *n* = 4, *p* < 0.01). Altogether, present results indicate that the reduction in effector caspase-3/7 expression induced by miR-138-5p overexpression reduces caspase activity and favors C6 and hippocampal neuron survival under apoptotic conditions.

### 3.7. Expression of Caspases 3 and 7 Increases after Subacute SCI and Negatively Correlates with the Downregulation of miR-18-5p

From the results described in the previous headings, it can be hypothesized that the downregulation of miR-138-5p in spinal neurons after SCI may cause the upregulation of caspase 3, caspase 7, and Bak1, increasing their sensibility to noxious stimuli. In order to evaluate the effects of miR-138-5p downregulation in the injured spinal cord on the expression of these pro-apoptotic targets, we measured their expression in spinal cord samples from a rat model of moderate contusive injury sampled 3 and 7 dpi. RT-qPCR analyses confirm the previously described [7] overexpression of Casp3 (*p* < 0.001), Casp7 (*p* < 0.05 and *p* < 0.01), and Bak1 (*p* < 0.05) at both 3 and 7 dpi (see Appendix A online). In agreement with the changes observed at mRNA level, protein levels of pro-CASP-3 (0.20 ± 0.07 for 0 dpi; 0.72 ± 0.20 for 3 dpi, 0.85 ± 0.48 for 7 dpi) and pro-CASP-7 (0.35 ± 0.17 for 0 dpi; 0.88 ± 0.32 for 3 dpi, 1.11 ± 0.47 for 7 dpi) in these samples also increase after injury (*p* < 0.001) (Appendix A online) as also does the Caspase-3/7 activity at 7 dpi (1.62 ± 0.09 normalized to 0 dpi, *p* < 0.01, see Appendix A online). Comparison with the miR-138-5p expression measured in the same samples reveals that the downregulation of this microRNA negatively correlates with the gene expression of its targets (Spearman’s Rs for Casp3 = −0.364, *p* = 0.136, Rs for Casp7 = −0.509, *p* = 0.055, Rs Bak1 = −0.524, *p* = 0.092, see Figure 8A) as well as with the caspases’ protein expression (Rs for pro-CASP-3 = 0.619, *p* = 0.051, Rs for pro-CASP- = 0.690, *p* < 0.05, see Figure 8B).

Since our initial analyses demonstrated that expression of miR-138-5p occurs mainly among spinal neurons (Figure 1), which also show the most evident post-SCI changes in the expression of this microRNA (Figure 2), we carried out a preliminary exploration of whether neuronal expression of caspases 3 and 7 reflects the expression patterns of miR-138-5p. To this aim, we quantified the expression of caspases in ventral and dorsal horn neurons of naïve and injured spinal cords. As shown in Figure 8C, the intensity of CASP-3 staining in neurons from the naïve spinal cord is markedly higher in the dorsal horns than in the ventral ones (dorsal horns: 75.22 ± 41.75, ventral horns: 45.52 ± 29.78), just opposite to the expression patterns observed for miR-138-5p. Such a negative correlation would suggest a negative post-transcriptional regulation of Casp3 by miR-138-5p. On the contrary, CASP-7 expression does not show horn-associated differences in neuronal expression (dorsal horns: 59.33 ± 31.14, ventral horns: 57.41 ± 22.50), which would suggest a lack of regulation of this protein by miR-138-5p in spinal neurons.

After SCI, CASP-3 staining in the spinal neurons at the injury penumbra (1 cm surrounding the injury epicenter) shows a general and progressive increase at 3 and 7 dpi (0 dpi: 63.67 ± 38.17, 3 dpi: 74.81 ± 49.08, 7 dpi: 81.63 ± 50.44; Figure 8D). It may be tempting to relate such an increase in expression with the overall downregulation of miR-138-5p shown in Figure 2 as a result of the demonstrated regulation of Casp3 by the microRNA. On the contrary, the intensity of CASP-7 staining in neurons after injury does not show any clear relationship with miR-138-5p expression changes, increasing at 3 dpi but reversing to control values at 7 dpi (0 dpi: 57.05 ± 24.78, 3 dpi: 73.11 ± 31.43, 7 dpi: 50.78 ± 24.71).

## 4. Discussion

We designed the present study to search for evidence of the participation of miR-138-5p downregulation in the cell death processes that characterize SCI pathophysiology. MicroRNAs are known to regulate the identity, state, and fate of neural cells [15]. Not surprisingly, microRNA dysregulation accompanies multiple neurological pathologies [49], including SCI [50]. Focusing on microRNAs that could contribute to cell death progression during SCI pathophysiology, we identified miR138-5p as a potential candidate due to its dysregulation after SCI [12,14] and its targeting of the SCI-upregulated pro-apoptotic protein CASP-3 [25].

Previous studies have shown that this CNS-enriched microRNA [46,51] is highly expressed among brain neurons [22,52,53,54,55]; however, to our knowledge, this is the first time that the cellular expression of miR-138-5p has been analyzed in the spinal cord. Our analyses in rats (*Rattus norvegicus* Wistar strain) detected miR-138-5p expression in nearly 60% of spinal neurons. Positive neurons are unevenly distributed across the spinal cord cross-section, being 30% of the dorsal horn neurons (Rexed laminae I to IV) and above 70% of the neurons in all other laminae (Figure 1E). A similar distribution is observed in the spinal cord of mice (*Mus musculus* C57BL/6J strain) according to ongoing studies from the lab (see Uceta [56]). In fact, regionalization of neuronal expression of miR-138-5p is also observed in the brain [46,55,57] and seems common among microRNAs (see, for example, miR-34 expression in spine neurons in Chang and cols. [58]). Besides neurons, miR-138-5p staining is also observed in APC-marked oligodendrocytes, though with lower intensity and without any clear sign of regionalization. The expression of this microRNA has been previously described in CNS oligodendrocytes from the central cerebellar region and corpus callosum from P17 mice (see Appendix A of Dugas and cols. [59]), where it is thought to promote the early stages of oligodendrocyte differentiation while preventing their final maturation, thereby facilitating the correct selection of axons and appropriate myelination [59]. No staining was detectable among GFAP-stained astrocytes in agreement with observations in the brain by Daswani and cols [57].

Following spinal cord injury, our RT-qPCR analyses in a rat model of moderate contusion revealed that miR-138-5p expression in the spinal cord becomes significantly reduced during the first week after contusion (3 and 7 dpi). This result agrees with previous observations of miR-138-5p downregulation in SCI [12,14,60] as well as in patients and animal models of various CNS injuries [61,62,63,64]. In contrast, Chen and Quin [65] described the upregulation of miR-138-5p in the rat spinal cord sampled 12 h after injury, which would suggest that miR-138-5p downregulation during the first week after SCI may be preceded by a short upregulation during the first hours. To understand the cellular basis of these bulk changes in miR-138-5p expression, we carried out histological analyses of spinal cord sections sampled 3 and 7 days after injury. The results indicate that the decrease in miR-138-5p expression at 3 dpi is produced from the loss of neurons and oligodendrocytes during the first days after injury. No changes in the percentage of miR-138-5p positive neurons and oligodendrocytes were apparent, so we can assume that the death of these cells is independent of miR-138-5p in these early stages. Interestingly, the liberation of the cellular contents from dead neurons and oligodendrocytes may explain the increase in plasma levels of miR-138-5p observed in SCI patients [65] and raises the possibility of employing the levels of circulating miR-138-5p as a biomarker to assess neuronal and oligodendroglial death. At 7 dpi we observed a reduction in the percentage of miR-138-5p positive neurons, which, coupled with the preservation in the number of surviving neurons, suggests that miR-138-5p becomes downregulated in these cells. Such a neuronal response to SCI disturbances agrees with Bicker and cols.’s [52] observation that neuronal expression of miR-138-5p is highly responsive to pathologies and activity changes. Conversely, no changes in miR-138-5p expression are observed among oligodendrocytes, which show a similar percentage of stained cells before and after injury. Altogether, histological data reveal that the miR-138-5p downregulation observed during the first week after injury by bulk gene expression methods (RT-qPCR, microarray, RNA sequencing) is caused by two different processes, the death of neurons and oligodendrocytes during the first days and later by the downregulation of the miRNA in neurons.

MiR-138-5p downregulation in neurons from the damaged spinal cord may have important physiological consequences. Changes in neuronal expression of miR-138-5p can affect dendritic spine size and synaptic transmission, learning and memory performance, cell migration, or axon growth and regeneration [52,66]. Moreover, expression levels of miR-138-5p determine neuronal fate according to direct and indirect evidence from animal models of ischemic stroke and neuroinflammation [61,64,67]. Indeed, the reduced levels of miR-138-5p observed in patients with hippocampal sclerosis associated with mesial temporal lobe epilepsy [68] have led to speculation that “a precise regulation of miR-138 and accordingly its targets is involved in cellular processes that protect neurons from disturbances in network activity” [52]. In agreement with this hypothesis, our in vitro analyses indicate that overexpression of miR-138-5p rescues neurons from glutamate excitotoxic stimulation (LGA treatment) as well as protects C6 neural cells from STS and ETO stimuli. Therefore, the observed miR-138-5p downregulation in spinal neurons following SCI may sensitize them to the SCI deleterious environment. A sensitizing effect was previously observed in rats under LPS-induced proinflammatory conditions, in which miR-138-5p downregulation leads to increased apoptosis of hippocampal neurons [67], and it is also associated with increased neuronal dismissal in a rat model of ischemia/reperfusion injury [64]. However, Tian and cols. [61] reported that overexpression of miR-138-5p was deleterious, whereas miR-138-5p knockdown was neuroprotective in cultures of hippocampal neurons as well as in a rat model of ischemia/reperfusion injury. The deleterious effects of miR-138-5p in vitro are particularly striking because Tian and cols. [61] employ the same primary cultures of hippocampal neurons employed here, with the only difference that we employed 18 days old (E18) rat embryos to obtain the tissue for the cultures while Tian and cols. employed newborn rats (P0) that were further cultured for 7 days before functional experiments were carried out. It has been described that mir-138-5p increases its expression during neuronal development in vitro [69] opening the possibility that the differences in the responses to miR-138-5p upregulation between neuronal cultures by Tian and cols. [61] and ourselves are indeed identifying a developmental-dependent effect. In this respect, it is worth mentioning that we opted to evaluate the neuroprotective effects of miR-138-5p overexpression instead of the effects of the actual downregulation observed after SCI due to the low levels of miR-138-5p in embrionary neurons [59,69], which precluded the use of a downregulation strategy.

Our in vitro analyses also revealed that miR-138-5p cytoprotection is associated with a reduction in the enzymatic activity of effector caspases and, consequently, to an attenuation of the execution of the apoptotic program. Apoptosis is a major mechanism of neural cell dismissal during the secondary damage of SCI [2] whose spread after injury can be under the regulation of miRNAs targeting pro- or anti-apoptotic genes, such as miR-21 [70] or miR-200c [71]. Therefore, we focused our search for targets underlying miR-138-5p blockage of apoptosis on the pro-apoptotic genes that become dysregulated after SCI. Our in silico analyses identified CASP-3, CASP-7, and BAK as potential or validated miR-138-5p targets among the pro-apoptotic genes becoming upregulated after SCI [7,9,35,53,54,72]. Reporter and immunoblot assays confirmed miR-138-5p post-transcriptional regulation of CASP-3 (in agreement with Chan and cols. [25]) and established for the first time miR-138-5p regulation of CASP-7 and BAK. CASP-3 and CASP-7 are cysteine-aspartic acid proteases which, upon activation, execute the apoptotic program and play a central role in the extension of apoptosis after SCI [73,74] whereas BAK is a member of the BCL2 protein family that contributes to the activation of apoptosis through the permeabilization of the mitochondrial outer membrane to release apoptogenic factors, including cytochrome c [48]. The gene expression of the three genes as well as the protein expression of the two caspases increases during the first week after SCI (present data, see also [7,54]) and negatively correlates with the expression of miR-138-5p, supporting their regulation by the miRNA. However evidence from the histological analyses of caspases in the neurons of naïve and injured spinal cords is less straightforward, supporting the regulation of CASP-3 particularly in the uninjured spinal cord but showing no relation between the expression levels of the microRNA and CASP-7 neither before nor after injury. Even though these analyses are preliminary and further analyses are required to confirm our results, they make clear that analyses at the cellular level are essential to fully understand how miR-138-5p downregulation affects the expression of both caspases and BAK after SCI.

To further complicate the scenario, other microRNAs are known to target these pro-apoptotic proteins, such as miR-106b and miR-337-3p on CASP-7 [75,76], miR-17-5p, miR-132-3p and miR-212-3p on CASP-3 [76], and miR-125b on BAK [77]. Interestingly, except for miR-106b and miR-337-3p, all these microRNAs appear dysregulated during the first week after injury according to our prior microarray analyses [14] and similar studies [12,70] and, therefore, may cooperate to regulate the apoptotic pathway in the damaged spinal cord. The scenario becomes even more complex when we consider the targeting of miR-138-5p on other cell death-related genes, such as BLCAP and MXD1 [25], LCN2 [78,79], BIM [80], Mst-1 [81], and Mlk3 [82,83], or even in the inhibition of the JNK and p38MAPK pathways [84]. Some of these targets—including Mst-1, BIM, and MLK3—contribute to the cell death processes of secondary injury [85,86]. Moreover, neuronal downregulation of miR-138-5p may also induce cell death indirectly due to the overexpression of its proliferative and cell cycle targets including EZH2, E2F2, E2F3, CDK6, CCND3, or ABL1 [87,88,89]. In fact, available evidence indicates that EZH2, E2F2, E2F3, and CCND3 become upregulated in the damaged spinal cord [90,91,92] and induce neuronal death during SCI’s secondary damage [9,90,93]. Additional effects are expected from miR-138-5p targeting additional genes, such as SIRT1 or MDR1, to name a few. Targeting the autophagy initiation regulator SIRT1 [94] has been proposed to contribute to the deleterious blockage of this process in neurons and oligodendrocytes following SCI (see Muñoz-Galdeano and cols. [95] and references therein) as well as to induce inflammation and apoptosis [65]. Meanwhile, miR-138-5p downregulation may also limit cytoprotective treatments due to its targeting of MDR1, a channel that confers drug resistance in cancer [96], which becomes upregulated in the damaged spinal cord to limit the effects of Riluzole [97].

## 5. Conclusions

We have shown that miR-138 downregulation during subacute SCI results from the initial loss of miR-138-5p expressing neurons and oligodendrocytes and the subsequent downregulation of this miRNA in neurons. Our results also show that miR-138 protects neurons and other neural cells from deleterious stimuli, and this microRNA can inhibit the execution of apoptosis in neural cells through the targeting and negative regulation of the pro-apoptotic proteins BAK, CASP-3, and CASP-7. Data from present and previous studies agree that downregulation of miR-138-5p after SCI can be deleterious to postmitotic neurons through a mixture of direct effects mediated by the upregulation of apoptotic targets and indirect effects related to the upregulation of cell cycle proteins. Further studies will be necessary to determine the precise contribution of miR-138-5p dysregulation to the activation of these cell death pathways after CNS injury. However, up to date, the available evidence suggests that miR-138-5p may be a valid neuroprotective target for SCI.

## Figures and Tables

**Figure 1 biomedicines-10-01559-f001:**
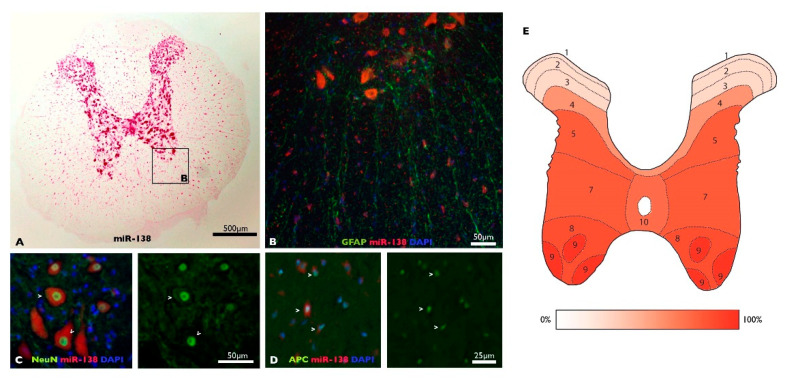
MiR-138-5p expression in the undamaged spinal cord: (**A**) Representative image of miR-138-5p expression (pink) in a transverse section of the thoracic spinal cord. (**B**–**D**) Co-staining of GFAP (astrocytes, (**B**), NeuN (neurons, (**C**) and APC (oligodendrocytes, (**D**) cell markers (green) with miR-138-5p probe (red) in spinal cord sections. (**E**) Map of the Rexed laminae in the T9 spinal segment detailing the mean percentage of miR-138-5p positive neurons present in each lamina of the naive spinal cord (7 sections from 2 individuals).

**Figure 2 biomedicines-10-01559-f002:**
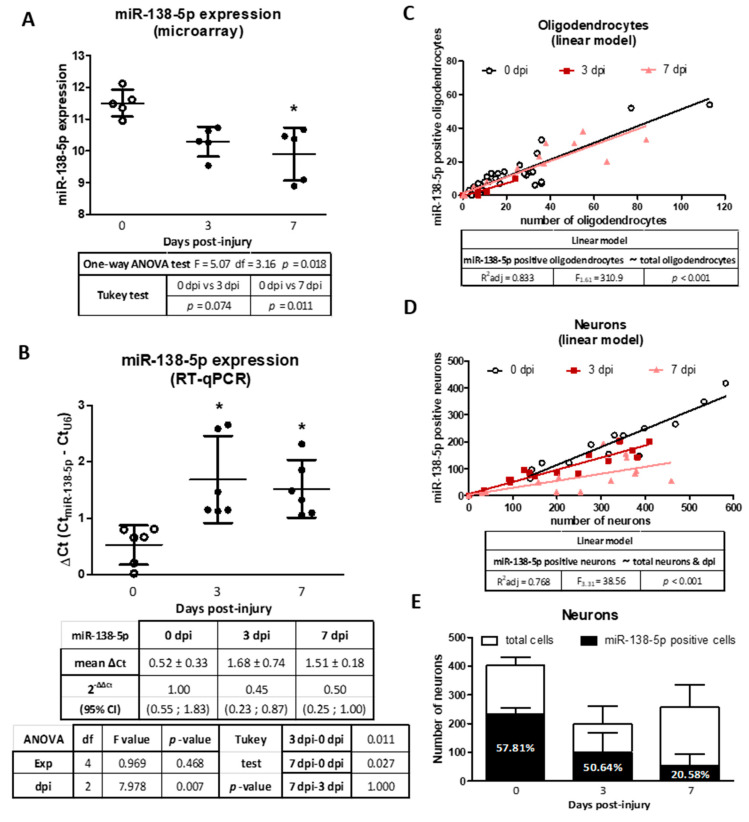
Changes in miR-138-5p expression at 3 and 7 days after SCI: (**A**) Spinal cord expression of miR-138-5p according to microarray data from Yunta and cols. (2012). The dot plot represents the average expression ± SD from five animals. (**B**) RT-qPCR showing the relative miR-138-5p expression in spinal cord samples collected at the indicated days post-injury. The graph represents the average of ΔCt from six independent experiments ± 95% CI. (**C**–**E**) Analysis of miR-138-5p expression in oligodendrocytes and neurons from FISH-stained slices of spinal cord sampled at 0, 3, and 7 dpi. The linear model adjusted in (**C**) relates the number of miR-138-5p positive oligodendrocytes in each section (dependent variable) with the total number of oligodendrocytes in the section (independent variable). Including the time after injury (dpi) in the model did not improve its adjustment and was discarded. (**D**) An equivalent linear model adjusted for the number of miR-138-5p positive neurons (dependent variable) indicates that it depends on the total number of neurons present and the day after injury (dpi). (**E**) Bar graph detailing the average number of neurons and miR-138-5p positive neurons per section in spinal cords from 3 animals per time after injury ± SD. * denotes *p* < 0.05.

**Figure 3 biomedicines-10-01559-f003:**
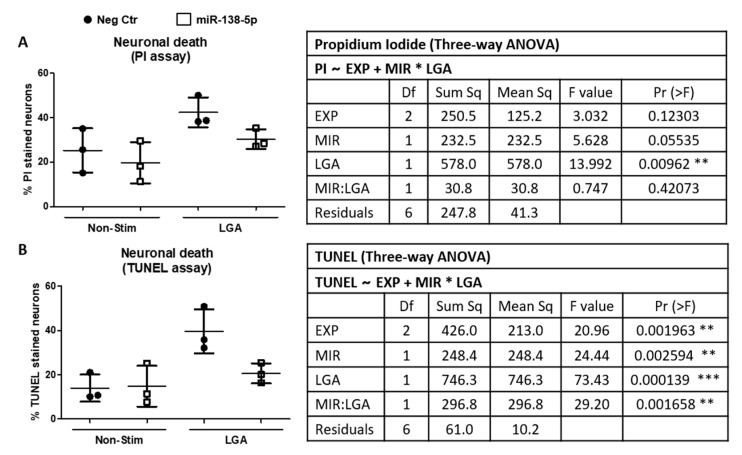
Effects of miR-138-5p transfection on neuronal death after LGA treatment. Neuronal death was analyzed in hippocampal neuron cultures transfected with either miR-138-5p (white squares) or negative control (Neg Ctr; black circles) mimics followed by LGA stimulation. (**A**) PI death assay. (**B**) TUNEL assay. Graphs represent the average of three independent experiments ± SD. The attached tables detail the results from the three-way ANOVA analyses. PI ~ EXP + MIR*LGA or TUNEL~EXP + MIR*LGA describes the employed ANOVA block design to determine whether PI or TUNEL variables depend on a block factor (experiment, EXP) and two independent variables (MIR for miR-138-5p transfection; and LGA for LGA stimulation) whose effects may depend on the values of the other independent variable (interaction term, MIR:LGA). *, ** and *** denote significant differences: *p* < 0.05, *p* < 0.01 and *p* < 0.001, respectively.

**Figure 4 biomedicines-10-01559-f004:**
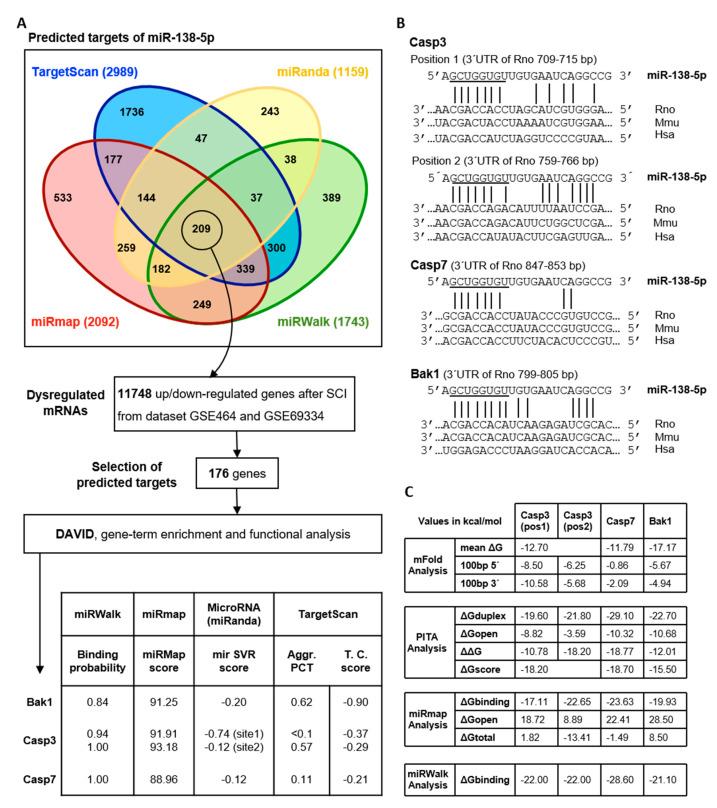
Bioinformatics analyses of miR-138-5p targets among pro-apoptotic proteins overexpressed after SCI: (**A**) Description of the analysis and major results in the identification of SCI-induced apoptotic targets of miR-138-5p. The upper Venn diagram shows the number of miR-138-5p target genes predicted by miRmap, TargetScan, miRWalk, and miRanda. 209 targets predicted by all four algorithms (overlapping areas of the four predictive tools) were compared with gene profiling data from rat models of SCI (GEO datasets GSE464 and GSE69334). The biological functions of the miR-138-5p targets undergoing expression changes after SCI were analyzed with DAVID functional annotation tools. The table shows the miR-138-5p predicted targets related to apoptosis and their prediction scores: (**B**) Alignment of the seed regions of miR-138-5p with the 3′UTR of Casp3, Casp7, and Bak1 in human (Hsa), mouse (Mmu), and rat (Rno). (**C**) Accessibility for the predicted targets according to mFold, PITA, miRmap, and miRWalk 3.0.

**Figure 5 biomedicines-10-01559-f005:**
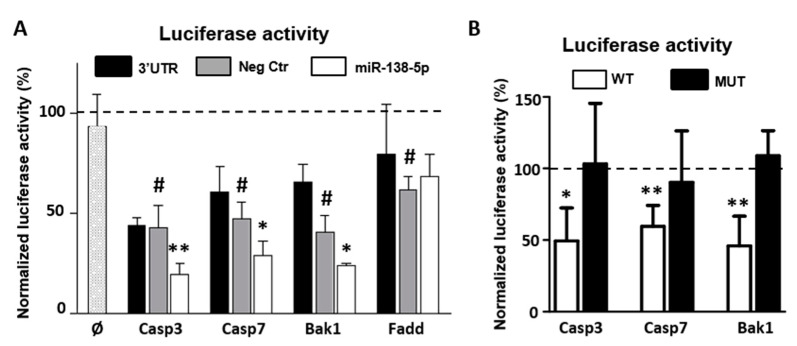
MiR-138-5p targets the 3′UTR of Casp3, Casp7, and Bak1: (**A**) Luciferase activity assays were carried out in C6 neural cell line cultures transfected with the complete 3′UTR of its three targets and co-transfected with mimics of miR-138-5p (white bars), negative control (Neg Ctr, gray bars), or without mimetics (black bars). Additional controls included C6 cultures transfected with a Luciferase construct without 3′UTR (Ø bar) or with Fadd 3′UTR. Values are expressed as the percentage of the luciferase activity of the empty construct (reference value represented as a dotted line in each experiment). Bars represent the mean ± SD of three independent experiments except for Casp3 (*n* = 5). Luciferase activity of the different 3′UTR reporter constructs co-transfected with the negative control microRNA mimic (Neg Ctr) was compared to the empty construct through a paired *t*-test (# denotes a significant difference: *p* < 0.05). The effect of transfecting miR-138-5p mimic was compared to the effect of the negative control (Neg Ctr) using a paired *t*-test (* and ** denote significant differences: *p* < 0.05 *p* < 0.01, respectively). (**B**) Luciferase assays carried out in C6 cell line cultures transfected with the wild type (WT; white bars) or mutant (black bars) 3′UTR of Casp3, Casp7, and Bak1. The dotted line represents the luciferase activity of the same construct transfected with Neg Ctr (set to 100 in each experiment). Bars represent the mean ± SD of four independent experiments. Luciferase activity of the WT and mutant 3′UTRs constructs co-transfected with the miR-138-5p mimic was compared to the values obtained after transfection with Neg Ctr using a paired *t*-test (* and ** denote significant differences: *p* < 0.05 and *p* < 0.01, respectively).

**Figure 6 biomedicines-10-01559-f006:**
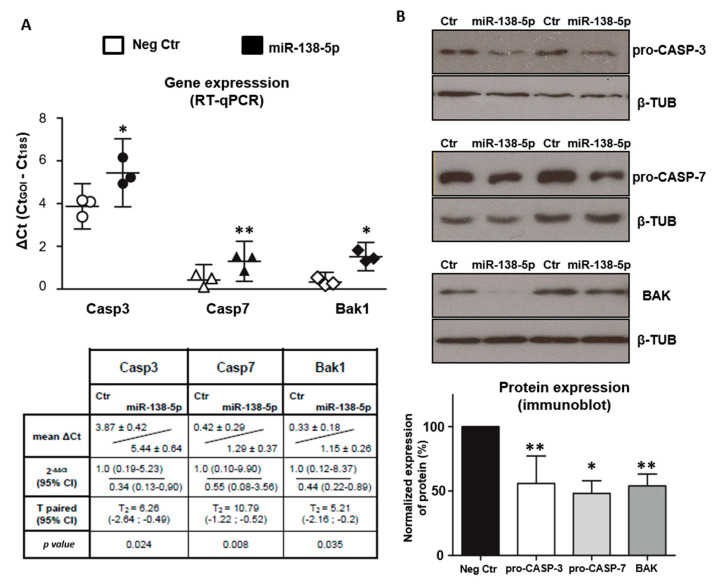
MiR-138-5p reduces the expression of Caspase 3, Caspase 7, and BAK: (**A**) Gene expression data from RT-qPCR analysis of C6 cultures transiently transfected with miR-138-5p or the negative control (Neg Ctr) mimics. The dot plot represents the average of ΔCt from six independent experiments ± CI. Comparison between expression values was carried out through a paired *t*-test with ∆Ct data (*n* = 3). (**B**) Analyses of protein expression of C6 cultures transiently transfected with miR-138-5p or Neg Ctr mimics and analyzed using immunoblot and densitometry. The images represent the results of two independent replicates. The bar graph underneath shows the densitometry data of each apoptotic protein following transfection with miR-138-5p relative to their respective values after transfection with Neg Ctr, and comparisons between the two were carried out through a paired *t*-test. The bars represent the average ± SD of, at least, four independent experiments. Complete blot images are available at OSF (https://osf.io/yqn58/, accessed on 24 May 2022). * and ** indicate *p* < 0.05 and *p* < 0.01, respectively, relative to its corresponding negative control.

**Figure 7 biomedicines-10-01559-f007:**
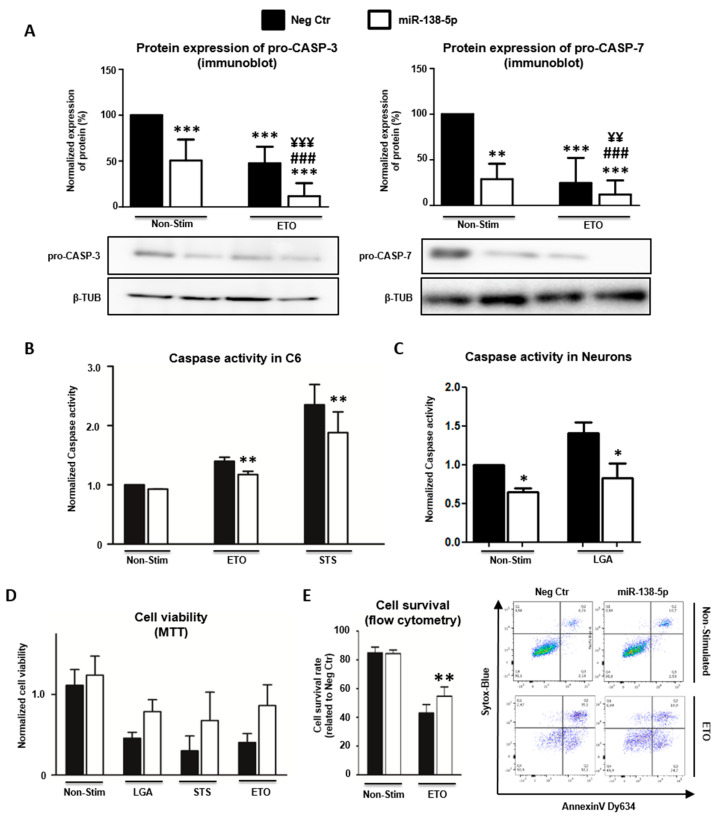
MiR-138-5p attenuates caspase-dependent apoptosis: (**A**) Protein expression of pro-CASP-3 and pro-CASP-7 in C6 cell cultures transiently transfected with miR-138-5p (white) or negative control (Neg Ctr, black) mimics and stimulated with ETO. Differences among conditions were calculated using the Kruskal–Wallis rank-sum test followed by Conover’s all-pairs post hoc test. ** and *** denote a significant difference (*p* < 0.01 and *p* < 0.001, respectively) relative to Neg Ctr; ### denote a significant difference (*p* < 0.001) relative to miR-138-5p-treated cultures; ¥¥ and ¥¥¥ denote significant differences (*p* < 0.01 and *p* < 0.001, respectively) relative to Neg Ctr + ETO-treated cultures. Bars represent the mean ± SD of, at least, four independent experiments. Complete blot images are available at OSF (https://osf.io/yqn58/, accessed on 24 May 2022). (**B**) Effects of miR-138 transfection on the activity of the effector CASP-3 and CASP-7 in C6 cell cultures before and after ETO and STS stimulation. Differences among conditions were calculated using a non-parametric Kruskal–Wallis test followed by a Conover post hoc test. Values correspond to the mean ± SD and ** denotes a significant difference (*p* < 0.01 and *p* < 0.001, respectively) relative to the corresponding cultures treated with the same stimulation and transfected with Neg Ctr. (**C**) Caspase activity in hippocampal neurons transfected with miR-138-5p or negative control before and after LGA stimulation. Differences among treatments were analyzed using a paired two-way ANOVA. Bars represent the average ± SD and * denotes a significant difference (*p* < 0.05) relative to the corresponding cultures treated with the same stimulation and transfected with Neg Ctr. (**D**) MTT assay of C6 cell survival after transfection with miR-138-5p and negative control mimics under different apoptotic stimulations. Differences among conditions were analyzed using a paired two-way ANOVA. Bars represent the mean ± SD of three independent experiments. (**E**) A representative flow cytometry experiment showing the survival rate of C6 cell cultures transfected with miR-138-5p or negative control mimics after 24 h of ETO stimulation. Cell cultures were co-stained with AnnexinV/SYTOX blue and analyzed using a flow cytometer. The associated graph summarizes the cell survival values (mean ± SD) of miR-138-5p or negative control mimic transfected cells. Differences among conditions were analyzed using a Kruskal–Wallis test and followed by a Conover post hoc test. ** denotes a significant difference (*p* < 0.01) relative to Neg Ctr + ETO-treated cultures.

**Figure 8 biomedicines-10-01559-f008:**
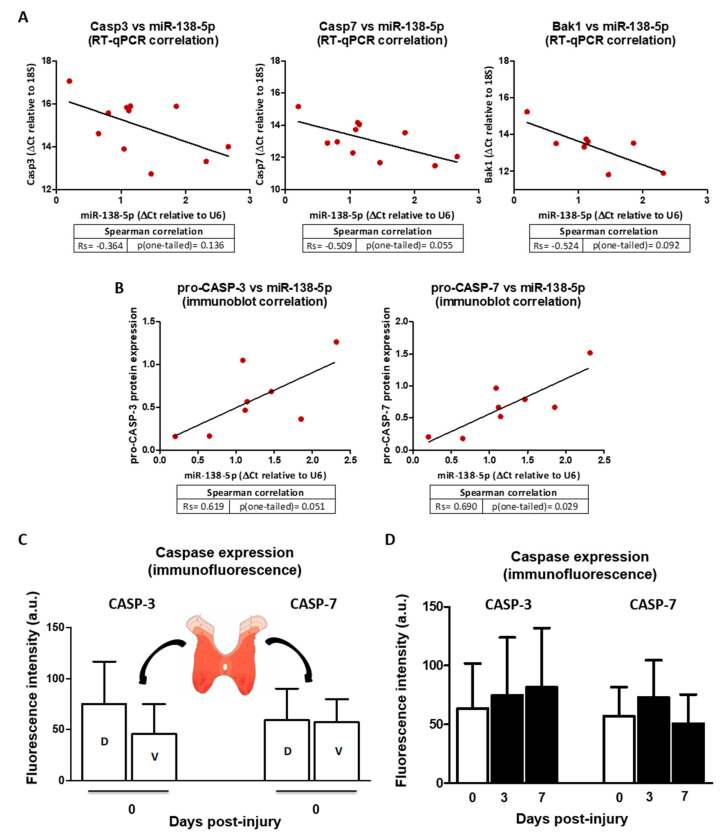
Relationship of miR-138-5p downregulation and caspase 3, caspase 7, and Bak overexpression after SCI: (**A**) Dot plots and regression lines represent the correlation between the expression of miR-138-5p and that of Casp3, Casp7, and Bak1. Samples from 11 individuals (*n* = 3 for 0 dpi; *n* = 4 for 3 and 7 dpi) were employed to analyze the correlation between caspases and miR-138-5p and from 8 individuals for Bak (*n* = 2 for 0 dpi, *n* = 3 for 3 and 7 dpi). (**B**) Dot plots and regression lines represent the correlation between miR-138-5p expression and pro-CASP-3 and pro-CASP-7 protein expression (the correlation appears as positive because ∆Ct values are inversely related to gene expression, whereas densitometric measurements of immunoblots are directly related to protein expression). Data employed correspond to 8 individual samples (*n* = 2 for 0 dpi and *n* = 3 for 3 and 7 dpi). (**C**) Analysis of CASP-3 and CASP-7 fluorescence expression in dorsal (D-labelled bars) and ventral (V-labelled) neurons from T10 segment of undamaged rat spinal cords. Bars represent the average ± SD of three animals. (**D**) Analysis of CASP-3 and CASP-7 staining intensity in spinal cord neurons of 0, 3, and 7 dpi. The graph represents the average ± SD of sections from three animals. All images are available at OSF (https://osf.io/yqn58/, accessed on 24 May 2022).

## Data Availability

All information obtained while conducting the present analysis is available at the Open Science Framework repository (OSF, https://osf.io/yqn58/ accessed on 24 May 2022).

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
