# Peer review of "MicroRNA-138-5p Targets Pro-Apoptotic Factors and Favors Neural Cell Survival: Analysis in the Injured Spinal Cord"

_biomedicines, 2022, doi:10.3390/biomedicines10071559_

Round 1
Reviewer 1 Report
The original article by Maza et al. "MicroRNA-138-5p targets pro-apoptotic factors and favors neural cell survival: analysis in the injured spinal cord" covers a potentially interesting and emerging topic related to the role of miRNA in pathogenesis of SCI. In this sense, this remains to be potentially interesting for the Biomedicines. I regard the main point of this paper as highly attractive as well as the results are clearly presented. The text does not contain any major errors, therefore I have some
minor comments and recommendations:
1. There is a need to provide slightly more expanded introduction shortly
mentioning/describing clinical course and global health problem of SCI
neuroinflammation such as traumatic brain injury (TBI), stroke, Parkinson's disease etc.
2. The figure summarizing and clarifying this schould be added to intoduction section
3. The potential use of anallyzed miRNA as the potential biomarker and
pharmacological target should be expanded
4. Following references should be added and properly cited within the main text:
- Correia de Sousa M, Gjorgjieva M, Dolicka D, Sobolewski C, Foti M. Deciphering miRNAs' Action through miRNA Editing. Int J Mol Sci. 2019 Dec 11;20(24):6249. doi: 10.3390/ijms20246249.
- Tykocki T, Poniatowski ŁA, Czyz M, Wynne-Jones G. Oblique corpectomy in the cervical spine. Spinal Cord. 2018 May;56(5):426-435. doi: 10.1038/s41393-017-0008-4.
- Sandrow-Feinberg HR, Houlé JD. Exercise after spinal cord injury as an agent for neuroprotection, regeneration and rehabilitation. Brain Res. 2015 Sep 4;1619:12-21. doi: 10.1016/j.brainres.2015.03.052.
- Wojdasiewicz P, Poniatowski ŁA, Turczyn P, Frasuńska J, Paradowska-Gorycka A, Tarnacka B. Significance of Omega-3 Fatty Acids in the Prophylaxis and Treatment after Spinal Cord Injury in Rodent Models. Mediators Inflamm. 2020 Jul 29;2020:3164260. doi: 10.1155/2020/3164260.
5. In some places the use of English could be improved on.
Completing this gaps will have an impact on the understanding the aim of the study and, from my point of view, is absolutely necessary.
Author Response
Asnwers to the Comments and Suggestions for Authors
- There is a need to provide slightly more expanded introduction shortly
mentioning/describing clinical course and global health problem of SCI
neuroinflammation such as traumatic brain injury (TBI), stroke, Parkinson's disease etc.
Answer: The main objective of our manuscript is to study the role of miR-138-5p on proapoptotic proteins and on neuroprotection after SCI. This objective is approached from a molecular and cellular point of view. On the other hand, this manuscript is part of a special issue on SCI, so we understand that there will be other articles that address this disease from its clinical course, or from the inflammatory pathway. Therefore, although we appreciate the suggestion, we believe that adding this information to the introduction section will distract the readers of the manuscript from the objective of this study. Anyway, we have included a short introductory sentence referring to the updated and well-documented review by Ahuja et al., 2017. This sentence indicates that: “Injury to the spinal cord (SCI) is a leading cause of permanent disabilities with dramatic physical, societal, and medical costs (for a detailed description see [1])”.
- The figure summarizing and clarifying this should be added to intoduction section.
Answer: thank you for the suggestion, but we do not consider a necessity or even an added value to summarize spinal cord injury pathophysiology or involved processes. There are several reviews that discuss SCI and the apoptotic pathway that we have cited in our article (e.g. Ahuja et al., 2017).
- The potential use of analyzed miRNA as the potential biomarker and
pharmacological target should be expanded.
Answer: As we mention in our conclusions “Further studies will be necessary to determine the precise contribution of miR-138-5p dysregulation to the activation of these cell death pathways after CNS injury.”Therefore, we are not really sure whether miR-138-5p may be a valid therapeutic target not to mention a valid biomarker. We feel that including such statements would be too speculative and would not add any value to the article. Analyses of gain and loss-of-function in vivo would be required to confirm the validity of miR-138-5p as a pharmacological target whereas a study specifically focused on its biomarker potential would be required for the latter.
- Following references should be added and properly cited within the main text:
- Correia de Sousa M, Gjorgjieva M, Dolicka D, Sobolewski C, Foti M. Deciphering miRNAs' Action through miRNA Editing. Int J Mol Sci. 2019 Dec 11;20(24):6249. doi: 10.3390/ijms20246249.
- Tykocki T, Poniatowski ŁA, Czyz M, Wynne-Jones G. Oblique corpectomy in the cervical spine. Spinal Cord. 2018 May;56(5):426-435. doi: 10.1038/s41393-017-0008-4.
- Sandrow-Feinberg HR, Houlé JD. Exercise after spinal cord injury as an agent for neuroprotection, regeneration and rehabilitation. Brain Res. 2015 Sep 4;1619:12-21. doi: 10.1016/j.brainres.2015.03.052.
- Wojdasiewicz P, Poniatowski ŁA, Turczyn P, Frasuńska J, Paradowska-Gorycka A, Tarnacka B. Significance of Omega-3 Fatty Acids in the Prophylaxis and Treatment after Spinal Cord Injury in Rodent Models. Mediators Inflamm. 2020 Jul 29;2020:3164260. doi: 10.1155/2020/3164260.
Answer: thank you for your suggestion. We agree with the referee that these articles are very interesting in their respective areas, but sincerely, we do not think any of these articles are really necessary for this article which is not focused on any of these topics.
- In some places the use of English could be improved on.
Answer: we have done our best to write this really lengthy manuscript. We feel that the English is good enough to be easily read and Grammarly tool did not yield any mistyping or wrong sentences. We agree that a review by a native English speaker would improve our writing but we do not have the funds for it.
Completing this gaps will have an impact on the understanding the aim of the study and, from my point of view, is absolutely necessary.
Answer: Thanks for your suggestions. We have done our best
Reviewer 2 Report
Authors of this study have investigated role of miR-138-5p after SCI, and they found that the SCI-induced downregulation of miR-138-5p may have deleterious effects on spinal neurons. The conclusion by the authors comes from extensive studies, and the findings are of clinical importance for understanding SCI. Overall, the manuscript is well written, but it still contains some minor points to be improved.
1) Contamination of SE and SD should be avoided in the entire manuscript. Please use either of which.
2) Decimal points. For example, 75.22±41.75 in line 760 should be 75.2±41.8. Please modify these points in the entire manuscript.
3) In abstract, authors described “We hypothesize that miR-138-5p 16 downregulation after SCI leads to overexpression of pro-apoptotic genes, sensitizing neural cells to noxious stimuli”. But this hypothesis has not been clarified. Did authors examine nociceptive sensitization by behavioral test? or electrophysiology? Please reconsider the hypothesis, if not.
Author Response
Answers to Comments and Suggestions for Authors
1) Contamination of SE and SD should be avoided in the entire manuscript. Please use either of which.
Answer: We agree with the referee that there were data expressed with Mean ± SEM and others expressed with Mean ± SD. To give coherence to the whole manuscript, we have changed the graphs of all the figures and now all are expressed with Mean ± SD. We have changed the figures embedded in the manuscript and their figure captions, and we have also attached them in a ZIP file containing both the figures and the supplementary material.
2) Decimal points. For example, 75.22±41.75 in line 760 should be 75.2±41.8. Please modify these points in the entire manuscript.
Answer: We have used 2 decimal points throughout all the text. Therefore, we have reviewed all the results in the manuscript and have changed and expressed all to 2 decimal points (e.g., 87.92±15.38), which has brought consistency to the text.
3) In abstract, authors described “We hypothesize that miR-138-5p 16 downregulation after SCI leads to overexpression of pro-apoptotic genes, sensitizing neural cells to noxious stimuli”. But this hypothesis has not been clarified. Did authors examine nociceptive sensitization by behavioral test? or electrophysiology? Please reconsider the hypothesis, if not.
Answer: Although we agree with the referee that the word sensitizing can refer to the effect on the nociceptive system and can be confusing, however, the medical dictionary (e.g. Thesaurus) indicates it also refers to the pharmacological effect of a drug, for example, the effect of miR-138-5p on the cellular state. Therefore, we believe that the phrase is correct and, therefore, the hypothesis is indeed answered by the experiments performed.